



**Vegetation Cover Change Detection and Assessment in Arid Environment Using**
**Multi-temporal Remote Sensing images and Ecosystem Management Approach**
Anwar Abdelrahman Aly[1, 3], Abdulrasoul Mosa Al-Omran*[1], Abdulazeam Shahwan Sallam[1], Mohammad
Ibrahim Al-Wabel[1], Mohammad Shayaa Al-Shayaa[2]
[1]Soil Science Dept., King Saud University, Riyadh, Saudi Arabia
[2]Agricultural Extension and Rural Community Dept., King Saud University, Riyadh, Saudi Arabia
[3]Soil and Water Science Dept., Faculty of Agric., Alexandria University, Egypt
**\***Corresponding Author: Tel: +966114678444; Fax: +966114678440
Email: rasoul@ksu.edu.sa; anwarsiwa@yahoo.com
**Abstract**
Vegetation cover (VC) changes detection is essential for a better understanding of the interactions and
interrelationships between humans and their ecosystem. Remote sensing (RS) technology is one of the most
beneficial tools to study spatial and temporal changes of VC. A case study has been conducted in the agro-
ecosystem (AE) of Al-Kharj, in the centre of Saudi Arabia. Characteristics and dynamics of VC changes
during a period of 26 years (1987 - 2013) were investigated. A multi-temporal set of images was processed
using Landsat images; Landsat4 TM 1987, Landsat7 ETM+ 2000, and Landsat8 2013. The VC pattern and
changes were linked to both natural and social processes to investigate the drivers responsible for the
change. The analyses of the three satellite images concluded that the surface area of the VC increased by
107.4% between 1987 and 2000, it was decreased by 27.5% between years 2000 and 2013. The field study,
review of secondary data and community problem diagnosis using the participatory rural appraisal (PRA)
method suggested that the drivers for this change are the deterioration and salinization of both soil and
water resources.  Ground truth data indicated that the deteriorated soils in the eastern part of the Al-Kharj
AE are frequently subjected to sand dune encroachment; while the south-western part is frequently
subjected to soil and groundwater salinization. The groundwater in the western part of the ecosystem is
highly saline, with a salinity $\geq 6$ dS m$^{-1}$. The ecosystem management approach applied in this study can be
used to alike AE worldwide.
**Keywords:** Change-detection, Remote sensing, Vegetation cover, PRA method, Al-Kharj agro-ecosystem

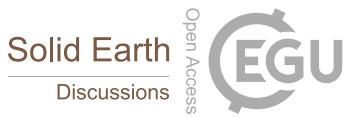

**List of abbreviations**
(EM) ecosystem management; (RS) remote sensing; (GIS) geographic information systems; (GPS) global
positioning systems, (LC) land cover; (LU) land use; (VC) vegetation cover; (HA) holistic approach; (AE)
agro-ecosystem; (PRA) participatory rural appraisal
**1. Introduction**
Many researchers working in ecosystem management (EM) find necessary to put communities as part of
ecosystem rather than treating them as separate entity (Aly, 2007; Reed et al., 2009). The ecosystems give
humankind many services such as provisioning services i.e., food, water, timber, fiber, and genetic
resources, regulating services i.e., the regulation of climate, floods, disease, and water quality, cultural
services i.e., recreational, aesthetic, and spiritual benefits, and supporting services i.e., soil formation,
pollination, and nutrient cycling (Bochet, 2015; Aly, 2007). Soil and vegetation as a part of ecosystems
give also many services to the humankind and play an important role in the earth system. The soil can act
as a filter of heavy metals and parasitic microorganisms; consequently, prevent plant and groundwater from
contamination (Keesstra et al., 2012; Brevik et al., 2015). Implementing sustainable EM implies improving
the quality of community life without depleting the ecosystems for future generations. Maltby (2000) and
Brodt et al. (2011) said that the newer concept of sustainability includes three dimensions, defined by three
broad goals: economic opportunity, social equity, and environmental health. When these goals are reached,
the sustainability will be achieved. However, Richardson et al. (2010) concluded that severe degraded
ecosystem may shift the EM goals from ecosystem restoration and sustainability to reconstructing entirely
new ecosystem. Since late 1980s an integration between EM, RS, GIS, and GPS has received substantial
consideration in the literature (Trabaquini, et al., 2012; Ehlers et al., 1989; Hinto, 1996). This integration
helps tackled more research problems related to EM. Nevertheless, the approaches by which these
techniques are integrated have become more complicated (Gao, 2002). Indeed the RS, GIS, and GPS are
providing desired technologies for land and environmental management (Seelan et al., 2003; Zucca et al.,



2015; Leh et al., 2015). Two terms are usually used in abundance by land management researchers, LC and
LU. The LC is defined as a physical material covered earth surface; however, LU is the human activities or
economic functions related to specific part of land (Singh, 2013). The LC comprises vegetation, asphalt,
bare ground, rivers, lakes…etc. Whilst the VC include only planted land i.e., grass, trees…etc (Singh,
2013; Aly, 2007). Loss of VC and plant species diversity reduces resistance of soil erosion and soil fertility
(Berendse et al., 2015; Yu and Jia, 2014; Cerdà and Doerr, 2005). The VC improve the infiltration rate and
decrease surface runoff and erosion (Cerdà, 1999). Furthermore, the VC have considerably affected the
global warming process through emissions of $CO_2$. However, C sequestration by afforestation in terrestrial
ecosystems could contribute to the decrease of atmospheric $CO_2$ rates (Muñoz-Rojas et al., 2015). The
analysis of the impact of LU changes on landscape processes can aid on the future policies of AE (Debolini
et al., 2015). The RS technology is usually used in EM (Mohawesh et al., 2015; Gong et al., 2015; Almeida
et al., 2005; Xie, 2008; Rawat, 2013; Croft et al., 2012). Vrieling (2006) concluded that four types of
factors are discussed by RS: topography, soil properties, VC, and management practices. Aly (2007) used
the RS technology in the HA of Siwa, located in Egypt, AE sustainable management. Furthermore,
Setiawan and Yoshino (2012) compared series of images through time to derive the land changes. Often
remote sensing imagery is imported into GIS software to facilitate analysis (Fichera et al., 2012).
Chowdary et al. (2001) used the Indian remote sensing satellite (IRS) data of 1988 and 1996 to monitor the
land resources and evaluate the land cover changes through a comparison of images acquired for same area
at different times. Yang and Yang (1999) analyzed different temporal images of 1996 TM 1992 TM, 1988
TM, 1982 MSS and 1979 MSS in purpose of detecting the coastal line change of Yellow River Delta.
Suliman (2001) acquired three different dated satellite Thematic Mapper images (TM) for 1984, 1993, and
1999 in addition to topographic maps to obtain new vulnerability map that can detect erosion, reclamation,
and development of Rosetta and Mutubas districts (markazes). El-Bana (2003) used two different dated
satellite TM images to obtain quantified changes in LU in northwestern part of Kafr El-Sheikh
Governorate, Egypt. Furthermore, Aly (2007) used three satellite images 1973 (MSS), 2000 (ETM), and





2005 (ASTER) to detect changes of LC in Siwa oasis, Egypt. Desprats et al. (2014) used satellite remote
sensing to identify VC in western part of Kingdom of Saudi Arabia (KSA). The use of RS and field studies
in the KSA summarized that sand dunes and soil and groundwater deterioration are considered the main
problems threaten the AEs (Aly et al., 2015; Algahtani et al., 2015; Alyemeni, 2000). The sand dunes cover
more than quarter of KSA surface (Alyemeni, 2000). These include four major sandy deserts (Nafud,
Dahna, Rub AI-Khali and Juffarah) in addition to other locally scattered sandy areas (Alyemeni, 2000). The
AEs is rarely found in vast dry land of KSA; furthermore, these AEs were usually considered fragile (Al-
Omran et al., 2014). Al-Kharj is a productive AE set in a desert depression in central of KSA and is
irrigated by waters originating from natural springs and dug wells with the lush of date palms, other fruits
(e.g. grapes), and vegetables (e.g. lettuce, carrots, tomatoes, cucumbers, and melons). It is a dryland fragile
AE that has a low degree of resilience to external stresses, and has a low carrying capacity (Al-Omran et
al., 2014). Some primary studies recorded that the soils and groundwater in Al-Kharj were deteriorating in
alarming way to lower suitability classes or sometimes to become unsuitable for cultivation (Al-Harbi,
2005). Consequently, the main objectives of this study are: i) to define the Al-Kharj, Saudi Arabia, AE
problems and sustainability using community diagnosis and field study ii) to detect the Al-Kharj's VC
changes using RS.   iii) Develop interventions that help restore the ecosystem's functions and integrity and
thus enhance the community's livelihood and promote social equity.
**2. Materials and methods**
**2.1 Study area**
The Al-Kharj is a fragile dryland AE has low resilience and carrying capacity. The ecosystem is located in
arid conditions in the middle of the Kingdom of Saudi Arabia (KSA) east of Riyadh city. It is set at 24°8′54″
N, 47°18′18″ E (Fig. 1). The groundwaters are considered the main source of irrigation, and the AE plants
various fruits and vegetables (e.g., date palms and grapes, and tomatoes, cucumbers, melons…etc.) (Al-
Omran et al., 2013). The Al-Kharj is located at 1360 m above sea level and its area is about 20.000 km $^2$ and has a
population of more than 600,000 people. There are only two large towns in the studied AE (Dilam and Asseeh);




however, there are three small towns (Al-Hayathim, Yamamah, and Sulamiyya). Furthermore, The AE include many
small hamlets and villages (Hagras et al., 2013). The Wadi (valleys) of Al-Kharj is discharged by water from Wadi
Hanifa and some other small wadis compensating part of consumed groundwater. The Al-Kharj include numerous
springs since ancient times; consequently, considered richest ecosystem in water resources in the KSA. The studied
AE has supported the KSA with grain, dairy products and other produced crops and livestock products. Recently, the
springs of Al-Kharj have dried up dramatically, like those in other places of the kingdom recurring drought
(McLaren, 2008).
**2.2 Ecosystem-Problem Identification (Community)**
The purpose of this part of the fieldwork is to identify the human activities and practices of the region,
particularly those that enhance ecosystem degradation within socio-economic and cultural constructs
(Swallow et al., 2009). The knowledge, attitude and practices (KAP) study was conducted using the
participatory rural appraisal (PRA) method, which includes the review of earlier study, field observation,
substantial indicators, town-hall meetings with community, sequence of one-on-one meetings, and build up
questionnaires. A town-hall meeting was held in Al-Kharj including around 250 persons of all stakeholders
and farmers. The questionnaire was field-tested, and modifications was made based on the results. The
most suitable format appears to be an easy-to-respond, non-time-consuming 'tick box' structure. To this
end, a suitable 123 questionnaire was designed collectively by the research team in consultation with the
local community to gather field information (Aly, 2007, Reed et al., 2009). Coding for different variables has
been accomplished and information gathered through the administered 123 questionnaires has been
statistically analyzed and the tasks accomplished is recorded in this study.

**Figure 1.** Study area location





### 2.3 Remote Sensing (RS): Change detection of the vegetation cover

RS by satellite images has been used since 1972 by first satellite, Landsat1 (Dogci and Kusek, 2008). Due to vast studied area, the proposed methodology is based on the use of remote sensing data. The very low cloud coverage on the Arabian Peninsula allows the acquisition of a global imaging cover the study area several years. In this study, a multi-temporal set of RS data of the Al-Kharj AE has been used to investigate vegetation cover changes (Fichera et al., 2012; Yuan et al., 2005; Lucas, 2007). The main parts can be distinguished by satellite image is the irrigated crops (Fig. 2). Three Satellite images over a period of twenty six years were acquired as follows:

1. Landsat4 TM: acquisition date is (27-11-1987), with seven spectral bands including thermal band. The ground sampling interval (Pixel size): 30 m reflective, 120 m thermal and scene size: 170 Km² X 185 Km² (Fig. 2a).

2. Landsat7 ETM+ : acquisition date is (16-12- 2000), with eight spectral bands , one of these bands is 15 m resolution in Panchromatic, 60 m thermal, and 30 m  other reflective bands (Fig. 2b).

3. Landsat8 : acquisition date is (28 -12-2013), with eleven spectral bands :

    - Multispectral bands 1-7,9 : 30 meters

    - Panchromatic band 8 : 15 meter

    - TIRS bands 10-11: resampled to 30 meter (Fig. 2c).

In order to mitigate the seasonal effects, which often lead to errors in change detection, the study adopted using only imagery acquired during the winter season, avoiding the uncertainness of inter-annual variability (Fichera et al., 2012).

**Figure 2.** Satellite images of Al-Kharj ecosystem



**2.4 Delineation of Vegetation cover**

In satellite images processing techniques, bands ratio usually represents special surface characteristics. The difference of two bands are called "index ". If this index comes from near Infrared to Red regions of spectral, it represents "Vegetation **index (VI)** ". The green plants have chlorophyll and reflect Infrared bands in high level; consequently, it appears in red color in the satellite images (GeoMart, 2011).

For the normalization of the vegetation index data, the vegetation **i**ndex has been divided by the total of the two bands. The result is then called "Normalized Difference Vegetation Index (NDVI) "and can be calculated as follow:

$$NDVI = (Nir - Red)/(Nir + Red)$$

The NDVI takes 32 bit data varying between (-1) and (1). The positive values represents the vegetation; however, the negative values represents the non-vegetated areas. These data can be scaled into 8 varying bit values (0 to 255). Where (-1) value goes to (0); on the other hand, (+1) value goes to (255). As a result of NDVI value, the light areas represent regions of high vegetation; however, the dark areas represent regions of low vegetation. This results can be extracted and masked in the pre-classification input data.

**2.5 Image Classification**

The three NDVI images obtained were classified in ERDAS software by the supervised classification as shown in Figure (3 A and B). The data of supervised classification calculates class means evenly and distributed in the data space then iteratively clusters the remaining pixels using minimum distance techniques. Each iteration recalculates means and reclassifies pixels with respect to the new means. This process repeated until the number of pixels in each class changes by less than the selected pixel change threshold or the maximum number of iterations is reached (Yuan et al., 2005; Lucas, 2007; Fichera et al., 2012). The overall accuracy values of each classified image are reported in Table 1.

The classified NDVI images were converted to vector layers (shape files) to detect and calculate the changes in the ecosystem vegetation cover (Fig. 4).



## 2.6 Field Study

### Water and Soil sampling and analysis

A 180 groundwater samples were gathered from different locations in the Al-Kharj AE to cover the spatial variations of the ecosystem groundwater salinity (Fig. 5). All samples were analyzed for salinity using EC meter (dS·m$^{-1}$) (Test kit Model 1500_20 Cole and Parmer) at 25 °C. The groundwater soluble calcium, magnesium, sodium, potassium, chloride, and sulfate were determined using Ion Chromatography System (ICS 5000, Thermo (USA)); however, the bicarbonate and carbonate concentration were determined by titration with sulfuric acid ($H_2SO_4$) (Matiti, 2004). Furthermore, fifty soil samples were collected from studied area including deteriorated sites observed by satellite image for year 2013 (ground truth). A soil paste extract were prepared, and the ECe was measured for each samples (Klute, 1986). In addition, A 5TE (Decagon devices) soil moisture, EC, and temperature sensors were installed at three field in the Al Kharj AE.

### Coordinate & GIS Analysis

In this study, the coordinates of the soils and groundwater samples were recorded by GPS with an accuracy of ~5 m. The GPS signal is corrected by a radio signal in real time. The locations of the ecosystem groundwater salinity were configured as a comma-delimited text file (in the form of groundwater no, easting, and northing). The point data was overlaid on a satellite image by Arc GIS 9.3 software (ESRI, 2010) (Fig. 5).

**Table 1** Accuracy assessment for the classified images.

**Figure 3.** NDVI classification for Landsat satellite image

**Figure 4.** Vector layer for classified NDVI

**Figure 5.** Location of the studied wells



## 3. Results and discussion

### 3.1 Community Diagnosis of Ecosystem Problems

Al-Kharj is a fragile ecosystem, highly vulnerable to environmentally induced land and water resources degradation. The ecosystem resource degradation problems in Al-Kharj are exacerbated by poor natural resource management and practices (Al-Omran et al., 2014).

The **PRA** approach was used in this study to undertake **diagnosis** of the community, and issues of land productivity and their importance as distinguished by the ecosystem community in order to interpret the changes found by RS (Shepherd, 2008; Mushove and Vogel 2005). The PRA is a concentrated, regular, but semi-structured learning practice conducted in a studied community by a multidisciplinary teamwork with complete contribution of the ecosystem community and stakeholders (Chambers, 1994; Mikkelsen, 1994). PRA help the researcher and community for identifying specific ecosystem problems and suggest solutions. The community diagnosis is considered a powerful investigation tool to overcome problems. The local people usually have an actual desire to solve their ecosystem problems. The PRA techniques were used to aid describe the issues that related to the characteristics of ecosystem to issues of agricultural and environment.

In this context, numerous appropriate PRA techniques were used specifically: the review of earlier study, field observation, substantial indicators, town-hall meetings with community, sequence of one-on-one meetings, and build up questionnaires. (Aly, 2007, Chambers, 1994).

As an important tool of the PRA methodology, a **general village- hall meeting** was organized and held in the ecosystem. The farmers are most of ecosystem residents but some of the inhabitants have other employments e.g. merchants, civil servants and labors. The meeting was carried out by a large number of the ecosystem community and stakeholders' e.g. local government administrators, and engineer and staff of agricultural extension. The meeting was managed to recognize how the community understand and prioritize their land and environmental deteriorations and issues facing them (Aly, 2007, Chambers, 1994).





Agricultural issues raised by PRA study
The PRA study found that the main agricultural problem in the studied ecosystem is the poor irrigation
water quality which is causing soil salinzation problems. This problem is aggravated by several factors
such as:
• The numbers of new wells that were recently drilled for irrigation has increased dramatically causing
depletion and deterioration of groundwater quality (Al-Omran et al., 2015).
• Poor irrigation practices in the Al-Kharj (excessive irrigation system).
• No agricultural drainage system in the Al-Kharj. Thus in some areas, the Al-Kharj could face the
danger of water logging and salinization problems.
• Large investments in intensive cultivation which cultivates hundreds of acres, and drilled tens of new
wells are causing great damage to the fragile ecosystem of the Al-Kharj.
• In summer there is no agriculture activity due to high temperature (reached 50ºC) with exception
protected area.
• Some farmer used desalination plant to overcome the irrigation water salinity.
• Loss of biodiversity due to soil salinity.
• Farmers in Al-Kharj usually change their soils when deteriorated.
• The rare and high cost of agricultural labor.
All the above mentioned agricultural issues and problems lead to a significant decrease in land productivity
of the studied agro-ecosystem.
**3.2  Remote Sensing: Direction changes of vegetation cover**
The major changes detected in the study area between years 1987 and 2000 were the increasing of VC in
west and south-western part of Al-Kharj ecosystem (Fig. 6). However, the VC decreased between years
2000 and 2013 in the east and south-western part of Al-Kharj AE (Fig. 6).  The investigation of the three
satellite images concluded that the surface area in square kilometers of the VC increased dramatically
between years 1987 and 2000 by 107.4%; however, it decreased by 27.5% between years 2000 and 2013





(Table 1) (Fichera et al., 2012). In an attempting to explicate the reason of the ecosystem VC decrease in last
decade, a relation between VC and wheat production has been depicted. Figure (7) shows a direct
relationship between wheat production in Saudi Arabia (USDA, 2015) and VC in Al-Kharj AE.
Furthermore, it recorded an evidence of progressive increase of wheat production and VC during the period
of 1984-1993 (USDA, 2015; Modaihsh et al., 2015; Algahtani et al., 2015). This was caused by the
economic development that corresponds to the period of massive injection of subsidies that came with
government's policy to expand the wheat production over this period (USDA, 2015). Rationally, this has
led to a steady increase in the land area used up by vegetation. However, there were a nosedived during the
period of 1994- 1998 due to the Saudi government stopped subsidies of wheat production to save water. A
slight increase of VC recorded between years 1998- 2002, and a contentious decrease between years 2002- 2013.
This study suggest that the decrease in the last decade of VC was caused by land and water resources
degradation (Sonneveld et al., 2016). This suggestion have been emphasized by field studies through PRA
method and found in agreement with the finding of Algahtani et al. (2015).
**3.3 Soil and water resources characteristics and its effects in agro-ecosystem**
The field study and observation, the review of secondary data, and community problem diagnosis using the
PRA suggest that the driving role in the change of VC recorded by RS in recent years are the soil and water
resources deterioration and salinization. The ground truth found that the deteriorated soils are either
subjected to salinization or sand dune encroachment (Fig 8, 9 and 10) (Alyemeni, 2000; Al Omran et al.,
2015). In general, the sand dune in eastern part of studied AE is considered the main problem facing
agriculture expansion; however, the groundwater salinity is considered the main problem of southwestern
part (Fig 8, 9 and 10) (Al Omran et al., 2015).
Table (2) shows that in the eastern part of the ecosystem 83% of groundwater samples were suitable for
irrigation with some restriction (ECw≤ 3 dS m$^{-1}$) (Ayers and Westcot, 1985); however, the remaining their

**Figure 6.** Change detection of vegetation cover

**Figure 7.** The changes of vegetation cover and wheat production





ECw ranged between 3-4 dS m$^{-1}$ (Table 2). In response to irrigation water salinity, 76% of irrigated soil
ECe ≤ 4 dS m$^{-1}$, 18% ECe  ranged between 4-10 dS m$^{-1}$, and 5% soil ECe >10 dS m$^{-1}$. Nonetheless, the VC
area decreased by 18% between years 2000-2013. In the middle and western part, the ecosystem showed
more vulnerable for degradation. Only 64% of the groundwater can be considered suitable for irrigation
(ECw≤ 3 dS m$^{-1}$). However, 20% of groundwater samples ECw ranged between 3-4 dS m$^{-1}$, and 16% the
ECw ranged between 4-10 dS m$^{-1}$. As a result, only 19% of the studied soil samples ECe≤ 4 dS m$^{-1}$, 50%
ECe between 4-10 dS m$^{-1}$, and regrettably 31% their ECe>10 dS m$^{-1}$. The VC is then decreased
dramatically in this part by 33% between the years 2000-2013. The highest soil ECe in eastern part of
studied ecosystem was 17.6 dS m$^{-1}$ (sample no 1); on the other hand, the middle part of the ecosystem
deteriorated sites recorded 40.6 and 47.4 dS m$^{-1}$, samples no 17 and 18, respectively (Table 3 and Fig. 10).
Moreover, the soil salinity dramatically increase in some sites of western ecosystem reaching 41.7 dS m$^{-1}$
(site no 29) (Fig. 6 and 10). The groundwater in western part of studied ecosystem is considered highly
saline since its salinity almost more than 6 dS m$^{-1}$ (Fig. 9). Mostly, no soil sodicity hazards are anticipated
by using this type of groundwater in irrigation. The SAR of studied waters were less than 10 with an
average of 3.74 (Table 4) (Richards, 1954). In general, 34.8% of the arable land in the studied AE are
considered saline (ECe > 4 dS m$^{-1}$), 34.8% are severely saline (ECe > 10 dS m$^{-1}$) and the remaining
(30.4%) can be considered non saline (ECe < 4 dS m$^{-1}$). The ECe of Al-Kharj cultivated soils are ranged
between 1 and 47.4 dS m$^{-1}$ for un-deteriorated and deteriorated sites, respectively; however, the
uncultivated soil's ECe reached 140 dS m$^{-1}$ in some sites.

**Table 2** Water and soil deteriorated parameter (salinity) and VC area

**Table 3** Statistical analysis of studied groundwater

**Table 4** Descriptive statistics of Al-Kharj groundwater

**Figure 8.** Sand dune encroachment

**Figure 9.** Interpolation of groundwater EC

**Figure 10.** Soil salinity in of studied ecosystem





### 3.4 Vegetation cover (VC) degradation and land and water resources salinity

In order to prove that the land and water resources salinity of past ten years are the main cause of VC

decrease in the ecosystem, the changes of VC has been linked to water and soil salinity levels (Fig. 11).

Three date palm fields with different changes of VC between years 2000 and 2013 were investigated. The

first field is located in eastern part of the study area with no change of VC and used fresh water for

irrigation (ECw = 1.1 dS m$^{-1}$). The second is a deteriorated field located in middle to the western part and

used saline brackish water for irrigation (ECw = 6.5 dS m$^{-1}$). The third is abandoned field located in

southern part of the study area with notable decrease of VC, this field has no irrigation activities due to the

high salinity of groundwater (ECw=10.2 dS m$^{-1}$) (Figs. 9 and 11). The first two irrigated fields adopted drip

irrigation system. A 5TE (Decagon devices) soil moisture, EC, and temperature sensors were installed at

each field. The average values of soil parameters (salinity, soil moisture and temperature) of four date

palms at depth (0-30 cm) for each field were presented (Fig. 11). The sensors in abandoned field did not

work properly due to the low soil water content (~ 0.01 m$^3$m$^{-3}$) where the precipitation is negligible (Gao et

al., 2014; Saha et al., 2015); therefore, the ECe (measured in saturated soil past extract) is presented (Fig.

11). The results indicated that the irrigation with low water salinity in the first field did not lead to high soil

salinity values (average soil's EC= 1.25 dS m$^{-1}$) (Fig. 11). The leaching process led to the soil salinity to

get lower with adding irrigation. However, the irrigation with saline water in the second field led to soil

quality deterioration due to salinity (average soil's salinity was equal to 6.7 dS m$^{-1}$) (Fig. 11). The soil in

the abandoned field is suffering from severe salinity (averaged 39.2 dS m$^{-1}$) due to lack of irrigation and the

low precipitation. Subsequently, soluble salts have been accumulated in the top soil layer negatively

impacting on VC water uptake and growth due to low tolerance of the VC to very high salinity. These are

expected results as salinization and alkalinization are the most common land degradation processes in arid

and semi-arid regions (Farifteh et al., 2006). Since the temperature of Al Kharj reaching 45 °C in July, the

soil temperature was also investigated in this study. Figure (11) clearly demonstrate that the summer

irrigation led to dramatic decrease of soil temperature (up to 5 °C). During the irrigation, the air is replaced



with water leading to the decrease in soil temperature. On contrary, following the irrigation, the water
drains and air would fill up the soil pores and the soil temperature gets higher (Fig. 11) (USAD, 2002).
Comparing the three site VC, it is clear that the high salinity of the land caused by high salinity of
groundwater resources had negative impact on vegetation survival especially in absence of leaching of salts
by rainfall or fresh water irrigation. In addition, the sand dune encroachment represent another cause of the
VC decrease in the eastern part of the study sites (Fig. 8). The farmers of Al-Kharj should be informed
about the water quality of their wells and should be given advice by the extension services about the type of
suitable crops and management that would safe guard the Al-Kharj ecosystem. The government should take
an action to solve the problem of sand dune encroachment in the eastern part of the ecosystem, and help
farmers to select salinity tolerance crops that can survive such conditions.
**Figure 11**. VC linked soil salinity
**4. Conclusions**
A comprehensive analyses of Al-Kharj, Saudi Arabia, agro-ecosystem components (physical resources and
community) were conducted in this study. The field study and community-based diagnosis in additions to
the use of satellite images to detect agriculture land-use changes over the twenty six years revealed that the
groundwater and agricultural lands have been seriously degraded due to salinization. The major ecosystem
changes detected by RS was VC surface area increased between years 1987 and 2000 by 107.4%; however,
it decreased by 27.5% between years 2000 and 2013. Between years 1984 and 1998, a direct relationship
between wheat production in Saudi Arabia and VC changes in studied AE is recorded. The Saudi
government subsidies to wheat production is governed the VC changes in this period. However, in the
following years, the degradation of land and water resources induced the VC changes. This study found
that the sand dune encroachment in eastern part of the AE is the main problem facing agriculture
expansion; however, the land and groundwater salinity is considered the main problem in the middle and
southwestern ecosystem. In the eastern ecosystem, 83% of the studied groundwater samples were suitable
for irrigation with some restrictions ($ECw \leq 3$ dS m$^{-1}$) and 76% of irrigated soil's $ECe \leq 4$ dS m$^{-1}$. However,



in the middle and western part, 64% of the groundwater can be considered suitable for irrigation (ECw$\leq$ 3
dS m$^{-1}$), and only 19% of the studied soil samples its ECe$\leq$ 4 dS m$^{-1}$. The farmers of Al-Kharj should be
informed about the water quality of their wells and should be given advice by the extension services about
the type of suitable crops and management that would safe guard the Al-Kharj ecosystem.
**Acknowledgment**
*This project was supported by NSTIP strategic technologies program number (12-ENV2581-02) in the*
*Kingdom of Saudi Arabia*

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






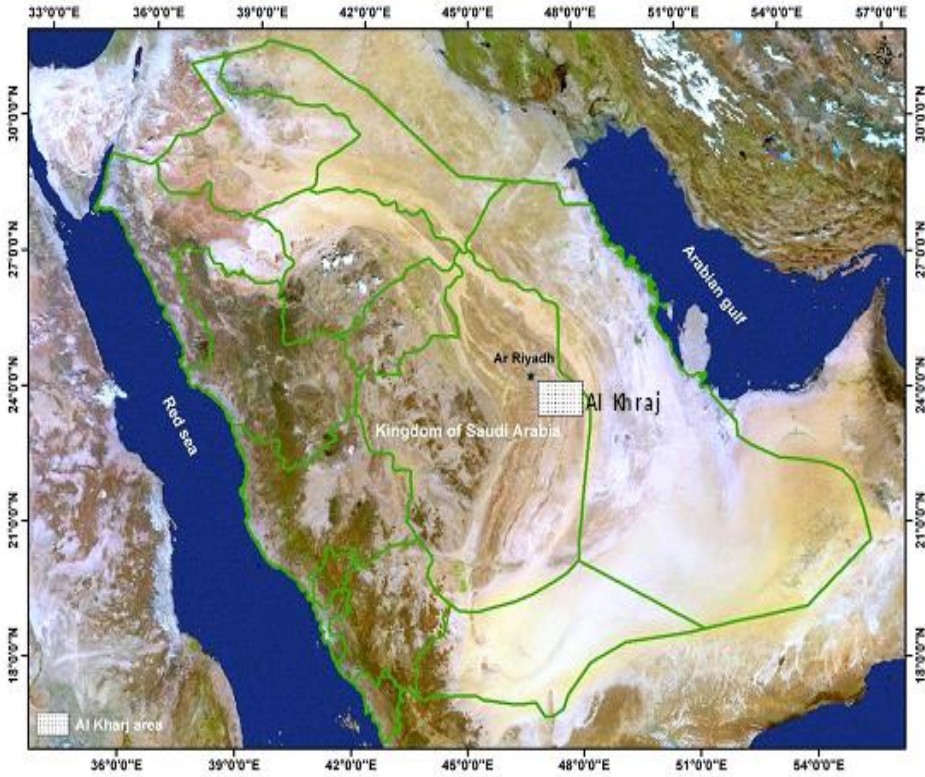

**Figure 1.** Location of the study area





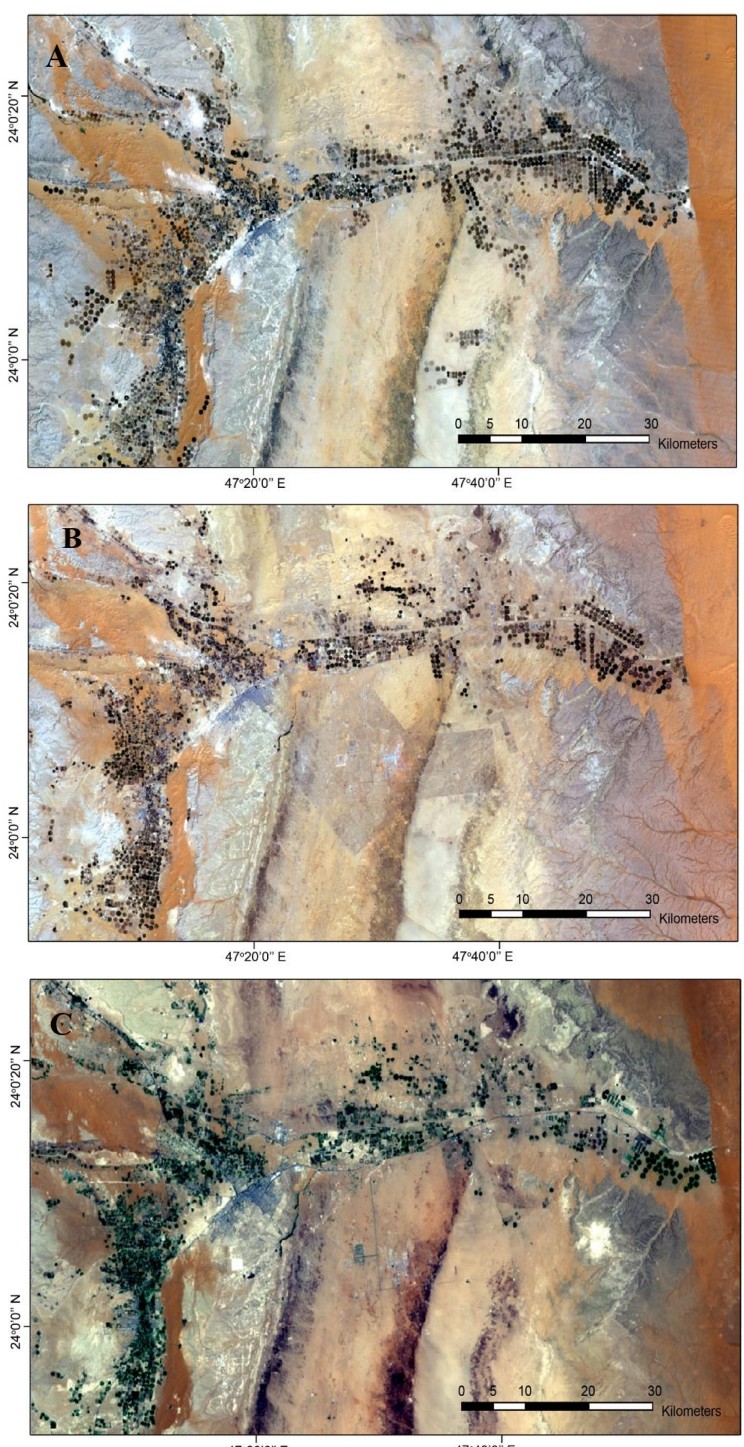

**Figure 2.** Satellite images of Al-Kharj ecosystem A) Landsat4 TM  B) Landsat7 ETM+
C) Landsat8






**Figure 3.** NDVI classification for Landsat satellite image of Al-Kharj A) 1987 B) 2013





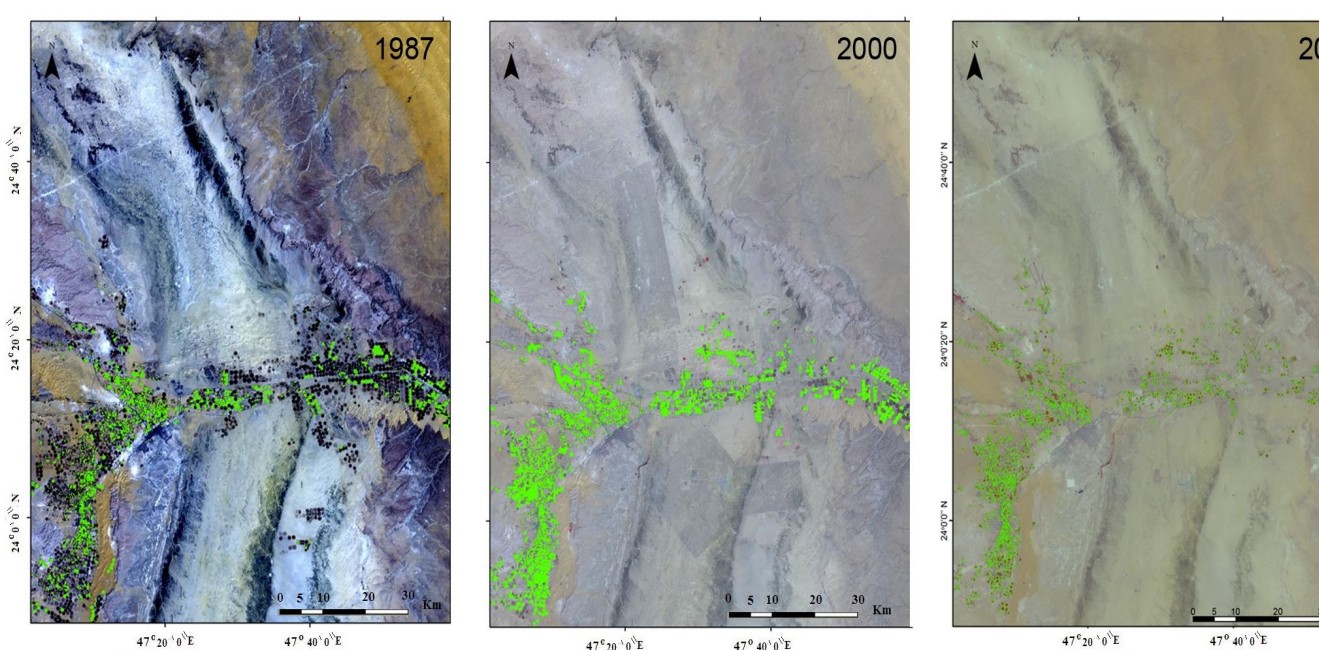

**Figure 4.** Vector layer for classified NDVI over Landsat satellite image 1987, 2000, and 2013 (Green color = cultivated area



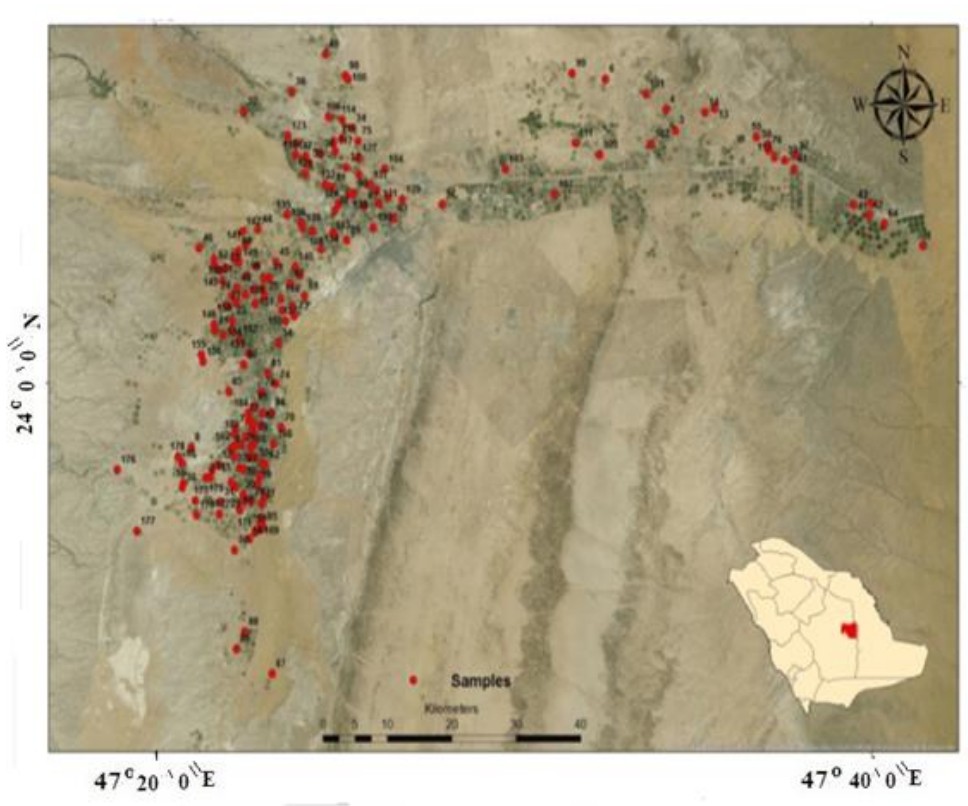

**Figure 5.** Location of the studied wells and soil samples





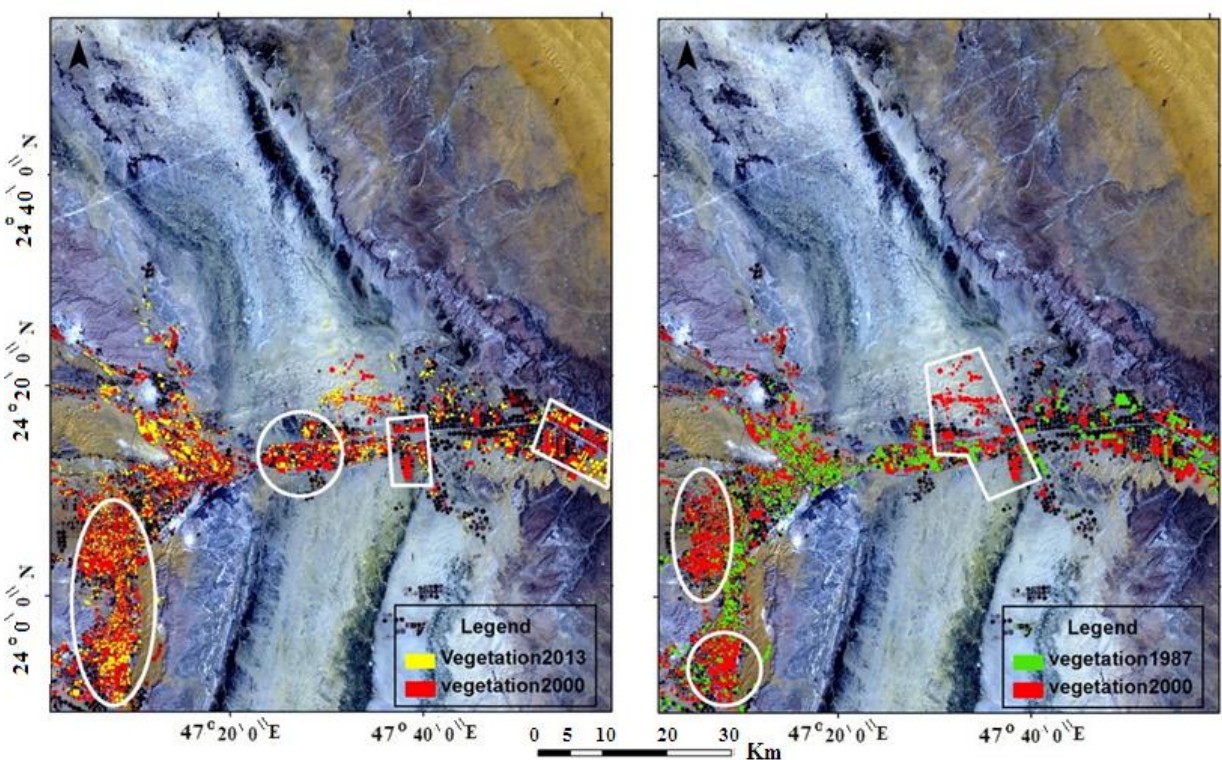

**Figure 6.** Change detection of vegetation cover: An increase observed between (1987 –2000) and a decrease between (2000-2013)





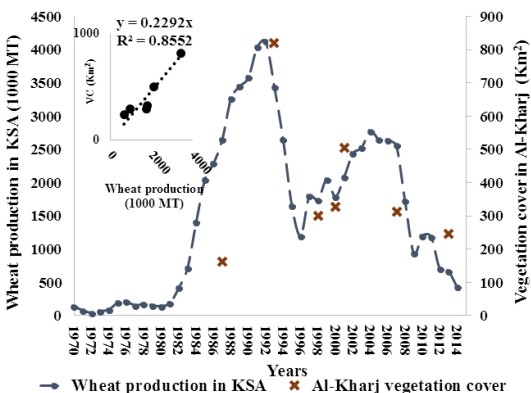

**Figure 7.** The changes of vegetation cover (VC) (km$^2$) and wheat production (1000 MT) of the Al-Kharj. The three RS date, 1993, 1998, and 2001, were for Landsat-5 cited by Modaihsh et al. (2015). The 2007 image was for Landsat Thematic Mapper (TM) cited by Algahtani et al. (2015)

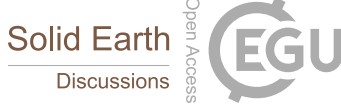



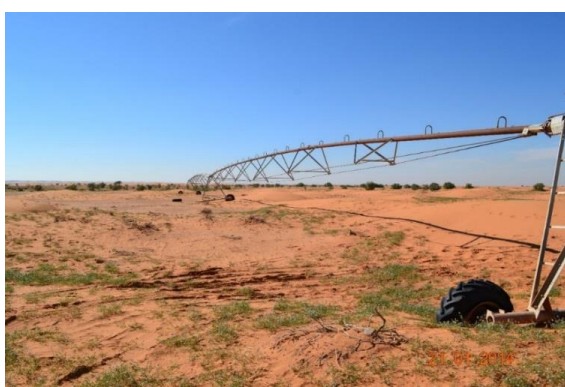

**Figure 8.** Sand dune encroachment in eastern part of Al-Kharj ecosystem





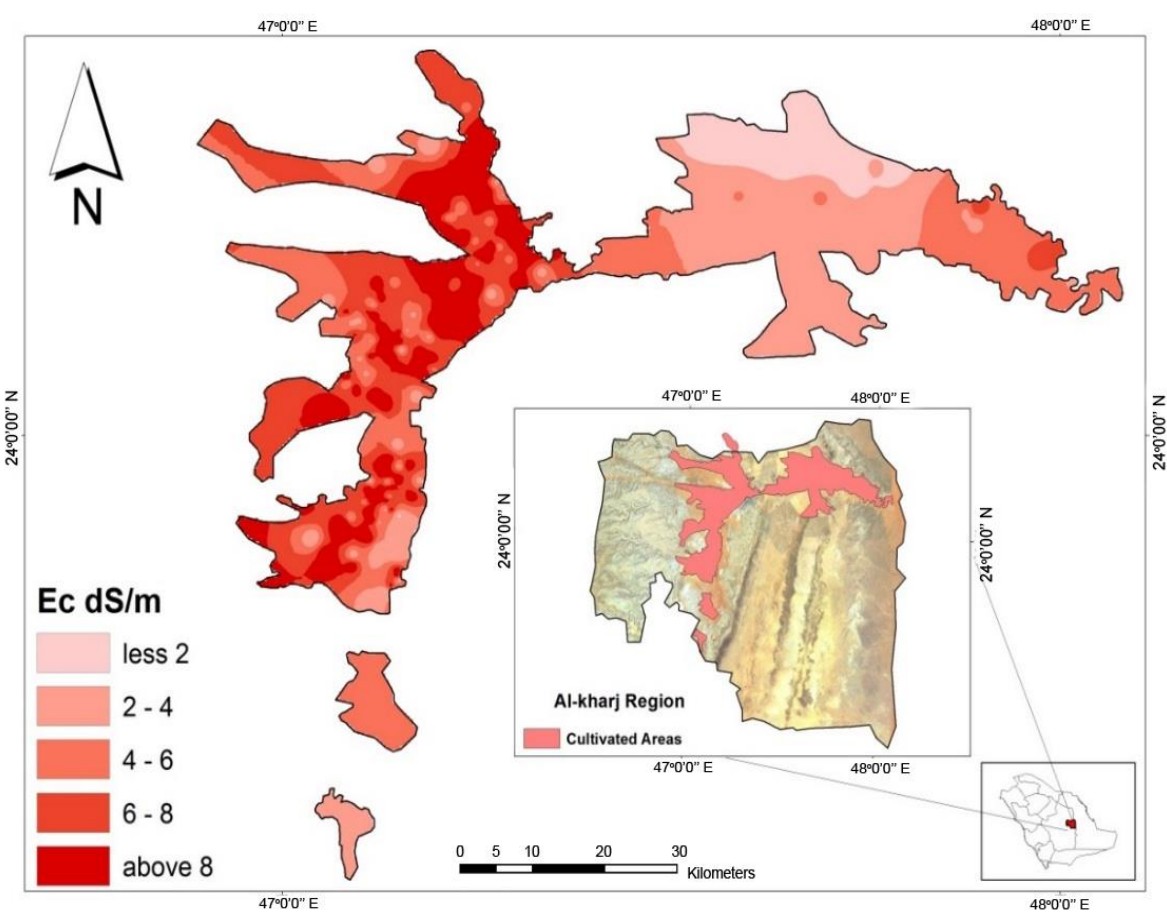

**Figure 9.** Interpolation of groundwater EC





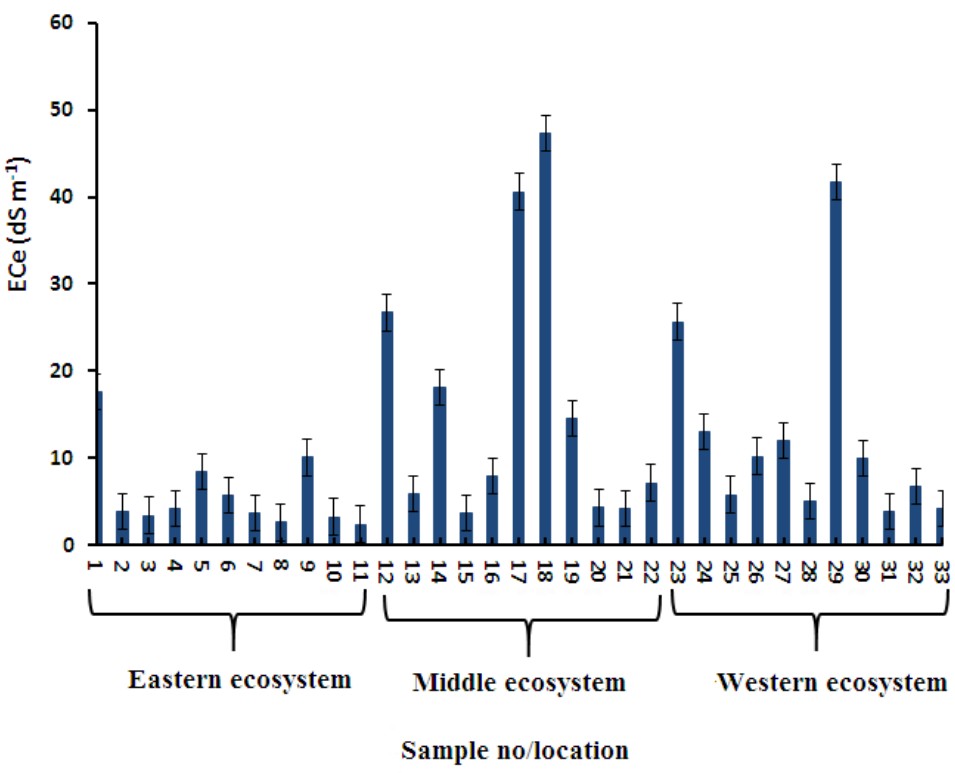

**Figure 10.** Soil salinity of some site of studied ecosystem



**Figure 11.** The changes of VC between years 2000 and 2013 linked to soil salinity, water content, and temperature at different fields in Al Kharj ecosystem; A) Field with no change of VC and used fresh water for irrigation (ECw = 1.1 dS m⁻¹).

B) Deteriorated field used saline brackish water for irrigation (ECw = 6.5 dS m⁻¹).

C) Abandoned field with no irrigation (ECw=10.2 dS m⁻¹).



**Table 1.** Accuracy assessment and ecosystem calculated areas of the classified images.

| Reference Year | Classified image | Overall Classification Accuracy | Ecosystem calculated area (Km$^2$) |
|---|---|---|---|
| 1987 | Landsat4 TM | 82.4% | 163 |
| 2000 | Landsat7 ETM+ | 86.5% | 338 |
| 2013 | Landsat8 | 96.1% | 245 |



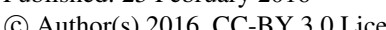



**Table 2.** Water and soil deteriorated parameter (salinity) in relation to VC area

| | | ECw[1] | | | ECe[2] | | |
|---|---|---|---|---|---|---|---|
| | | ≤ 3 | 3 - 4 | 4-10 | ≤ 4 | 4-10 | >10 |
| Eastern Ecosystem | % of samples | 83 | 17 | - | 76 | 18 | 5 |
| | VC % decrease (2000-2013) | | | 18 | | | |
| Middle and western Ecosystem | % of samples | 64 | 20 | 16 | 19 | 50 | 31 |
| | VC % decrease (2000-2013) | | | 33 | | | |

[1] ECw = The EC of water sample

[2] ECe = The EC of soil sample determined on soil paste extract (Klute, 1986).





**Table 3.** Descriptive statistics of soil and groundwater in ecosystem areas subjected to sand dune encroachment (eastern part) or salinization (middle and western part)

| | Soil | | Water | |
|---|---|---|---|---|
| | Eastern part | Middle and western part | Eastern part | Middle and western part |
| Max | 17.63 | 47.35 | 3.82 | 10.15 |
| Min | 2.50 | 2.34 | 1.31 | 1.83 |
| Mean | 3.05 | 12.11 | 2.50 | 3.22 |
| Median | 2.66 | 7.12 | 2.54 | 2.73 |
| St. deviation | 7.51 | 12.01 | 0.71 | 1.42 |




**Table 4.** Statistical analysis of groundwater chemical composition of Al-Kharj (n=180)

| | PH | EC dS m$^{-1}$ | Ca$^{2+}$ | Mg$^{2+}$ | Na$^+$ | K$^+$ | Cl$^-$ | HCO$_3^-$ | CO$_3^{-2}$ | SO$_4^{-2}$ | SAR |
|---|---|---|---|---|---|---|---|---|---|---|---|
| | | | | | | meq L$^{-1}$ | | | | | |
| Max. | 8.60 | 10.15 | 36.75 | 29.85 | 43.40 | 0.72 | 58.17 | 18.83 | 4.33 | 43.19 | 9.14 |
| Mini. | 6.78 | 1.05 | 3.45 | 0.79 | 2.24 | 0.05 | 3.13 | 0.87 | 0.00 | 3.22 | 1.08 |
| Mean | 7.72 | 3.00 | 10.79 | 7.78 | 11.28 | 0.25 | 10.86 | 3.99 | 0.13 | 15.03 | 3.74 |
| Stdev | 0.44 | 1.29 | 5.09 | 3.93 | 5.96 | 0.10 | 7.32 | 1.49 | 0.37 | 7.05 | 1.47 |
| Vari. | 0.66 | 1.13 | 2.26 | 1.98 | 2.44 | 0.31 | 2.71 | 1.22 | 0.61 | 2.66 | 1.21 |
| St. error | 0.18 | 0.23 | 0.33 | 0.31 | 0.34 | 0.12 | 0.36 | 0.24 | 0.17 | 0.36 | 0.24 |
| Med. | 7.72 | 2.64 | 9.60 | 6.69 | 10.21 | 0.23 | 9.50 | 3.83 | 0.00 | 12.83 | 3.51 |
| Skew | -0.15 | 2.47 | 1.39 | 2.16 | 2.53 | 1.66 | 3.85 | 5.96 | 8.20 | 1.18 | 1.12 |