# Peer review of "Vegetation Cover Change Detection and Assessment in Arid Environment Using Multi-temporal Remote Sensing images and Ecosystem Management Approach"

_Solid Earth, 2016_

## Referee Comment (RC1) · Anonymous Referee #1 · 15 Mar 2016

I thank to SE Editorial Board for the request to review the manuscript "Vegetation cover change detection and assessment in arid environment using multi-temporal remote sensing images and ecosystem management approach" (see below my review report).

Comments to the Authors The paper presents an interesting topic about land use/cover changes focused on the detection of spatial and temporal changes of vegetation cover using remote sensing technology with potential interest and importance to examine the linkage between land cover changes, natural processes and anthropogenic activities. A consistent work was done with different methodological approaches using the

participatory rural appraisal, fieldwork by collecting soil and groundwater samples for salinity analysis and processing Landsat images using GIS techniques. The paper is well written and structured. The objectives are clear and the results are well interpreted. Although the aims of the manuscript are studied in previous studies as it is reported in the introduction, the manuscript present a good approach to understand the main drivers responsible for the vegetation cover changes in Al-Kharj agroecosystems (Saudi Arabia) identifying the environmental problems in the study area during the period 1987 – 2013 mainly linked to soil and groundwater salinization and sand dune encroachment. However, some questions are needed to improve the manuscript. Please find below the review comments:

Specific comments: Lines 17 – 19 Please revise the sentence. "A multi-temporal set of images was processed . . .Landsat8 2013 to investigate the drivers responsible for the VC pattern and changes which are linked to both natural and social processes"

Line 51 become more complicated? Authors should provide more information about this statement

Lines 67 Please complete the sentence with the location of the study of Setiawan and Yoshino (2012)

Line 102 I suggest to include the population of the two mentioned large towns to include a more detailed information about the study area

Line 105 The authors could provide an estimation of the number of springs in order to provide the magnitude of this source of water

Line 180 using electrical conductivity (EC) meter

Line 186 Please include both abbreviations in the text distinguishing between EC measured on groundwater samples (ECw) in line 180 and EC measured on soil samples (ECe) since these abbreviations are used in lines 273 – 294. I suggest including the soil-water ratio of prepared saturated paste extracts in line 186.

Lines 207 – 212; Lines 220 – 221 Authors mentioned the purpose of using PRA approach and the characteristics of this method. This paragraph seems to correspond to Materials and methods.

Line 226 Revise sentence

Lines 227 – 242 This is a major result and it should deserve a bit more discussion. I might recommend the authors to include references, examples or data.

Line 268 Figure 8 What type of kriging did the authors selected to model the spatial distribution of EC?

Line 279 Please revise the sentence "the ecosystem showed more vulnerable soil conditions for soil degradation" / "the ecosystem was more vulnerable for land degradation"

Line 294 Please specify in which sites

Lines 304 – 311 This paragraph is a description of the three date palm fields studied that it should be included in Materials and method section.

Lines 310 – 311 "A 5TE (Decagon devices) soil moisture . . . at each field" is repeated in lines 187 – 188.

Line 314 "therefore the ECe (measured in saturated soil past extract) is presented (Fig.11)" can be removed

Lines 331 – 335 According to the objective iii) in line 93. The proposed interventions should be discussed in more detailed.

Figures and Tables

Figure 4 there is no a full view of the shape for the year 2013

Figure 5 sampling points of soil samples (n=50) can be included

Figure 8 replace less and above by "<, >" symbols

Figure 10 caption: Salinity of selected soil samples (n=33)

Table 3 descriptive statistics of EC (ds/m) of soil and water samples. Please include the number of soil samples (n=50) and groundwater samples (n=180) as in Table 4.

Table 4 replace Mini. by Min; use the same abbreviations for standard deviation and median in Tables 3 and 4 and place a table note with the abbreviations i.e. St.Dev: standard deviation Vari: variance? and Swe: skewness

Technical corrections

Please ensure that the references are listed first alphabetically and then chronologically (e.g. see lines 39, 49, 52, 57, 58, 63,131,209)

Line 100 delete "and"

Line 120 (Aly, 2007; Reed et al.,2009)

Line 169 A and B in lowercase letter and include A and B letters in Figure 3 as in Figure 2.

Line 186 "sample" instead of "samples"

Line 228 salinization

Line 339 in addition to

Line 263 findings

Lines 268, 271 and 287 Figs

Line 289 replace waters by water samples

Appendix Environment – replace "is" by "it"; Forget and Lebel, 2001.

---

## Referee Comment (RC2) · Anonymous Referee #2 · 24 Mar 2016

⇢ Section 2.3 (line 126) should be renamed to "Remote sensing images characterization". ⇢ Section 2.4 (line 153) should be renamed to "Delineation of Vegetation cover changes" ⇢ What is the need for section 2.5 "image classification" using supervised classification for NDVI images???. Converting NDVI images to classified images is a simple straight forward process that doesn't need a supervised classification process. ⇢ The section named "Coordinate & GIS Analysis" (line 189) has no GIS analysis at all, and the authors did not mention anything about which interpolation method they used to get the groundwater salinity distribution map. ⇢ The paragraphs at lines 217 and 220 should be moved to methodology section ⇢ The authors didn't

mention anything in the methodology about the statistical analysis of the data • On what basis did the authors classify the ecosystem into eastern, middle and western??? • Classification accuracy should be removed from table 1 • Figure 3: is missing the letters A and B for the images • Figure 5 is missing the distinction between soil samples and groundwater samples • What is the meaning of different white shapes in figure 6?? • What is the relation of figure 9 to figure 5, and how the polygon of the interpolated salinity is obtained?? • In figure 10, why the authors selected 33 samples to plot soil salinity instead of the total 50 samples??? • Figure 11 is not clear. What is the meaning of the white shapes, and do the letters A B and C fall inside the shapes or not????? •

---

## Author Comment (AC1) · 30 Mar 2016

Dear Prof. Artemi Cerdà

Editor, Journal of Solid Earth

We would like to thank you very much for useful comments and suggestions of our manuscript titled "Vegetation Cover Change Detection and Assessment in Arid Environment Using Multi-temporal Remote Sensing and Ecosystem Management Approach". We modified the manuscript according, and detailed corrections are listed below point by point: We look forward to your positive response. Sincerely, Corresponding author

[Figure]

E-mail: rasoul@KSU.EDU.SA or anwarsiwa@yahoo.com

Anonymous referee 1

Comment: Lines 17-19 Please revise the sentence. "A multi-temporal set of image was processed….landsat8 2013 to investigate the drivers responsible for the VC pattern and changes which are linked to both natural and social processes". Respond: The sentence is revised.

Comment: Line 51 become more complicated? Authors should provide more information about this statement. Responds: More information included.

Comment: Lines 67 Please complete the sentence with the location of the study of Setiawan and Yoshino (2012). Responds: Setiawan and Yoshino (2012) compared series of images through time to derive the land changes in Tsukuba, Japan.

Comment: Line 102 I suggest to include the population of the two mentioned large towns to include a more detailed information about the study area. Responds: The population of the two mentioned large towns is included.

Comment: Line 105 The authors could provide an estimation of the number of springs in order to provide the magnitude of this source of water. Responds: No accurate estimation available of the total number of springs in the studied ecosystem.

Comment: Line 180 using electrical conductivity (EC) meter. Responds: Revised.

Comment: Line 186 Please include both abbreviations in the text distinguishing between EC measured on groundwater samples (ECw) in line 180 and EC measured on soil samples (ECe) since these abbreviations are used in lines 273-294. I suggest including the soil-water ratio of prepared saturated paste extracts in line 186. Responds: Both abbreviations included in the list of abbreviations.

Comment: Lines 207-212 Lines 220-221 Authors mentioned the purpose of using PRA approach and the characteristics of this method. This paragraph seems to correspond

to Materials and Methods. Responds: Paragraph on line 207-212 is transferred to Materials and Methods; however, the sentence on lines 220-221 found suitable in its part due to consistent with the context of the following sentences.

Comment: Line 226 revise sentence. Responds: Revised to be "Agricultural problems summarized by community study".

Comment Lines 227-242 This is a major result and it should deserve a bit more discussion. I might recommend the authors to include references, examples or data. Responds: More discussion included.

Comment: Line 268 Figure 8 What type of kriging did authors selected to model the spatial distribution of EC? Responds: The kriging, geostatistical method, interpolation in Fig 9 was carried out using kriging interpolation tool of Geostatistical analyst in ArcGIS 9.3. This was included in the materials and methods, the part of Coordinate & GIS Analysis.

Comment: Line 279 Please revise the sentence "the ecosystem showed more vulnerable soil conditions for soil degradation" "the ecosystem was more vulnerable for land degradation". Respond: The sentence revised.

Comment: Line 294 Please specify in which sites. Responds: In western AE.

Comment: Lines 304 – 311 This paragraph is a description of the three date palm fields studied that it should be included in Materials and method section. Responds: The paragraph included in the Materials and method section.

Comment: Lines 310 – 311 "A 5TE (Decagon devices) soil moisture . . . . . .at each field" is repeated in lines 187 – 188. Responds: The paragraph was revised.

Comment: Line 314 "therefore the ECe (measured in saturated soil past extract) is presented (Fig.11)" can be removed. Responds: Removed.

Comment: Lines 331 – 335 According to the objective iii) in line 93. The proposed

interventions should be discussed in more detailed.

Responds: The following paragraphs were included to the manuscript:

Sand dune fixation is generally used to stop the dunes encroachment. Two methods are usually used; biological i.e., planting trees, shrubs and grasses species, and mechanical i.e., wooden sand fences and footpaths. Shelterbelt systems and afforestation, biological methods, using Atriplex spp., Acacia spp, and Casuarina spp were found efficient in stabilizing dunes in arid environment of Egypt, Senegal, and India (Draz et al., 1992; Kaul, 1985). In fact, the importance of the sand dunes fixation by afforestation is not only sand dune fixation but also can conserve arid ecosystem balance, and produce fuel and animals feed (Draz et al., 1992; Kaul, 1985). In the USA, Tunisia, and Egypt saline waters have been successfully used for long irrigation time. The crops grown using this water are cotton, sugarbeet, alfalfa, date palm, sorghum, barley, alfalfa, rye grass and artichokes (Rhoades et al., 1992). In Texas, USA, the saline groundwater (TDS = 2500 to 6000 mg/l) has been successfully used for three decades (Rhoades et al., 1992). The suitability of saline groundwater for irrigation should be assessed for specific conditions including; crops type, soil characteristics, irrigation methods, cultural practices, and climatic conditions (Minhas, 1996). Many rational management option of saline irrigation water have been currently in use, some of them are: cyclic strategy, which involves using non-saline water and saline water in a repeating sequence, blending strategy which involves blending (dilution process) fresh with saline water, rotation strategy which means irrigation with low-salinity water for salt sensitive crops in a rotation with saline water for salt-tolerant crops (Rhoades et al., 1992), planting salt tolerant crop varieties or genotypes / cultivars i.e., amaranth and quinoa which can survive under harsh conditions (Fghire et al., 2015; Pulvento, et al., 2015), and finally the use of computer model for assessing water suitability for crops production (Aly et al., 2015).

Figures and Tables

Comment: Figure 4 there is no a full view of the shape for the year 2013. Responds: Revised.

Comment: Figure 5 sampling points of soil samples (n=50) can be included. Responds: Included.

Comment: Figure 8 replace less and above by "<, >" symbols. Responds: You mean Figure 9, revised .

Comment: Figure 10 caption: Salinity of selected soil samples (n=33). Responds: Revised.

Comment: Table 3 descriptive statistics of EC (ds/m) of soil and water samples. Please include the number of soil samples (n=50) and groundwater samples (n=180) as in Table 4. Responds: Included.

Comment: Table 4 replace Mini. by Min; use the same abbreviations for standard deviation and median in Tables 3 and 4 and place a table note with the abbreviations i.e. St.Dev: standard deviation Vari: variance? and Swe: skewness. Responds: Replaced and abbreviations included.

Technical corrections:

Comment: Please ensure that the references are listed first alphabetically and then chronologically (e.g. see lines 39, 49, 52, 57, 58, 63,131,209). Responds: Revised accordingly.

Comment: Line 100 delete "and". Responds: Deleted.

Comment: Line 120 (Aly, 2007; Reed et al.,2009). Responds: ";" included.

Comment: Line 169 A and B in lowercase letter and include A and B letters in Figure 3 as in Figure 2. Responds: A and B in lowercase letters are included and A and B letters included in Figure 3.

Comment: Line 186 "sample" instead of "samples". Responds: Revised.

Comment: Line 228 salinization. Responds: Revised.

Comment: Line 339 in addition to. Responds: Revised.

Comment: Line 263 findings. Responds: Revised.

Comment: Lines 268, 271 and 287 Figs. Responds: Revised.

Comment: Line 289 replace waters by water samples. Responds: Replaced.

Comment: Appendix Environment – replace "is" by "it"; Forget and Lebel, 2001. Responds: Replaced.

Please also note the supplement to this comment:
http://www.solid-earth-discuss.net/se-2016-31/se-2016-31-AC1-supplement.pdf

[Figure]

[Figure]

**Figure 3.** NDVI classification for Landsat satellite image of Al-Kharj A) 1987 B) 2013

**Fig. 1.** Revised fig 3

[Figure]

**Figure 4.** Vector layer for classified NDVI over Landsat satellite image 1987, 2000, and 2013 (Green color = cultivated area)

**Fig. 2.** Revised fig 4

[Figure]

**Figure 5.** Location of the studied groundwater and soil samples, and investigated fields

**Fig. 3.** Revised fig 5

[Figure]

**Figure 9.** Interpolation of groundwater EC

**Fig. 4.** Revised fig 9

**Table 1.** Accuracy assessment and ecosystem calculated areas of the classified images.

| Reference Year | Classified image | Ecosystem calculated area (Km$^2$) |
|---|---|---|
| 1987 | Landsat4 TM | 163 |
| 2000 | Landsat7 ETM+ | 338 |
| 2013 | Landsat8 | 245 |

**Fig. 5.** Revised table 1

[revised manuscript text omitted]

---

## Author Comment (AC2) · 3 Apr 2016

Anonymous referee 2 Comment: Section 2.3 (line 126) should be renamed to "Remote sensing images characterization". Respond: Renamed.

Comment: Section 2.4 (line 153) should be renamed to "Delineation of Vegetation cover changes". Respond: Renamed.

Comment: What is the need for section 2.5 "image classification" using supervised classification for NDVI images??? Converting NDVI images to classified images is a simple straight forward process that doesn't need a supervised classification process. Respond: Thanks for comment; however, different methods are adopted in Remote Sensing (RS). In general, there are three main image classification techniques in RS: • Unsupervised image classification • Supervised image classification • Object-based image analysis The image classification uses the reflectance statistics of individual pixels (smallest image unit). Unsupervised and supervised image classification techniques are the most common approaches used in RS.

Comment: The section named "Coordinate & GIS Analysis" (line 189) has no GIS analysis at all, and the authors did not mention anything about which interpolation method they used to get the groundwater salinity distribution map. Respond: The kriging, geostatistical method, interpolation in Fig 9 was carried out using kriging interpolation tool of Geostatistical analyst in ArcGIS 9.3. This is included in the section of Coordinate & GIS Analysis.

Comment: The paragraphs at lines 217 and 220 should be moved to methodology section. Respond: The paragraphs moved.

Comment: The authors didn't mention anything in the methodology about the statistical analysis of the data. Respond: Statistical analysis of the data is included in the materials and methods.

Comment: What basis did the authors classify the ecosystem into eastern, middle and western??? Respond: According to local classification.

Comment: Classification accuracy should be removed from table 1. Respond: Removed.

Comment Figure 3: is missing the letters A and B for the images. Respond: A and B for the images included.

Comment: Figure 5 is missing the distinction between soil samples and groundwater samples. Respond: Soil samples locations included in figure 5.

Comment: What is the meaning of different white shapes in figure 6?? Respond:

Locations of the vegetation cover changes.

Comment: What is the relation of figure 9 to figure 5, and how the polygon of the interpolated salinity is obtained?? Respond The interpolation of groundwater EC carried out using the EC of actual samples located in Fig 5. The interpolation in Fig 9 was carried out using kriging interpolation tool of Geostatistical analyst in ArcGIS 9.3.

Comment: In figure 10, why the authors selected 33 samples to plot soil salinity instead of the total 50 samples??? Respond: Illustration 50 samples in one figure is unacceptable; furthermore, the samples of uncultivated soils have an EC up to 140 dS/m. (this is illustrated in the tables and discussion section).

Comment: Figure 11 is not clear. What is the meaning of the white shapes, and do the letters A B and C fall inside the shapes or not????? Respond: The white shapes means the locations of VC changes, and the letters A, B and C fall inside the shapes showing the location of investigated fields. This is also explained using figure 5.

---

## Referee Comment (RC3) · Anonymous Referee #2 · 7 Apr 2016

Dear respective author

Thank you for your comment about the types of image classifications and meaning of the pixel, it was very informative.

It is the first time in my life to see the application of image classification technique to NDVI image. I wont argue with you more about the subject, and whether it is correct or not. The end this, please cite as many research articles as possible which used supervised classification to classify NDVI single band image that has no spectral signatures at all, and I need you to tell me how many training sites did you use to perform such a

classification.

---

## Referee Comment (RC4) · Anonymous Referee #2 · 11 Apr 2016

Dear respective authors

Thank you for your discussions about the NDVI image.

After deep reviewing of the literature for NDVI calculation I suggest the following modifications for the remote sensing part. Since NDVI is a calculated index, you do not need to prove the accuracy of the results you get. This will increase the scientific value of your good research.

Remove the last sentence of section 2.4, and replace it with the following: • The NDVI images could be classified into three classes, namely dense vegetation cover
(NDVI > 0.5), moderate vegetation cover (NDVI 0.25 – 0.5), and sparse vegetation cover (NDVI < 0.25), as shown in Fig (4).

Section 2.5 should be removed completely

Section 2.6 becomes 2.5

N.B. you can choose different threshold values other than these values, according to the literature you have.

N.B. you can add the area of each vegetation class, for each satellite image date in a table to show the changes in the vegetation cover (you do not need to convert the NDVI data into vector format for further analysis).

• Regarding table 1, the title of the table should be renamed to "spatio-temporal characteristics of Al-Kharj ecosystem", which will have the areas of the agroecosystem corresponding to each satellite date. I hope these suggestions will of acknowledged by the authors.

Best Wishes

---

## Author Comment (AC3) · 11 Apr 2016

Dear respective Reviewer

Thank you very much for valuable comments

After deep discussion with co-authors, we found that the section 2.5 "image classification" can be revised as follow.

2.5 Accuracy assessment

180 training sites were recorded for field trip to evaluate the NDVI classification method

to the study area. The overall accuracy value were calculated using ERDAS software for the NDVI classified images (Figure 3 a and b). The accuracy of NDVI classification of 2013 image was found 96.1%. The NDVI images were converted to vector layers (shape files) in order to detect and calculate the changes in the areas of vegetation cover (Table 1, Fig. 4).

Thanks Again and Kind Regards

---

## Referee Comment (RC5) · Anonymous Referee #2 · 14 Apr 2016

Dear Respective authors

Thank you very much for responding to the comments. I wish you all the best

---

## Author Comment (AC4) · 14 Apr 2016

Dear respective Reviewer

Thank you very much for valuable comments, below you will find our responds to the comments point by point:

Comment: After deep reviewing of the literature for NDVI calculation, I suggest the following modifications for the remote sensing part. Since NDVI is a calculated index,

[Figure]

you do not need to prove the accuracy of the results you get. This will increase the scientific value of your good research.

Respond: Thanks for comment, the accuracy removed.

Comment: Remove the last sentence of section 2.4, and replace it with the following: The NDVI images could be classified into three classes, namely dense vegetation cover (NDVI > 0.5), moderate vegetation cover (NDVI 0.25 – 0.5), and sparse vegetation cover (NDVI < 0.25), as shown in Fig (4).

Respond: The last sentence removed and replaced with the required sentence.

Comment: Section 2.5 should be removed completely Section 2.6 becomes 2.5.

Respond: The section 2.5 removed and the section 2.6 became 2.5.

Comment: N.B. you can choose different threshold values other than these values, according to the literature you have.

Respond: Ok, thanks.

Comment: N.B. you can add the area of each vegetation class, for each satellite image date in a table to show the changes in the vegetation cover (you do not need to convert the NDVI data into vector format for further analysis).

Respond: Table 1 include the vegetation class (Table 1 included below)

Comment: Regarding table 1, the title of the table should be renamed to "spatio-temporal ' characteristics of Al-Kharj ecosystem.

Respond: Table 1 renamed

Thanks Again and Kind Regards.

Please also note the supplement to this comment:
http://www.solid-earth-discuss.net/se-2016-31/se-2016-31-AC4-supplement.pdf

[Figure]

[Figure]

**Table 1.** Spatio-temporal characteristics of Al-Kharj ecosystem

| Reference Year | Classified image | Vegetation cover areas (Km$^2$) | | | |
|---|---|---|---|---|---|
| | | Dense (NDVI > 0.5) | Moderate (NDVI = 0.25 − 0.5) | Sparse (NDVI < 0.25) | Total |
| 1987 | Landsat4 TM | 36 | 69 | 58 | 163 |
| 2000 | Landsat7 ETM+ | 8 | 156 | 174 | 338 |
| 2013 | Landsat8 | 6 | 91 | 148 | 245 |

**Fig. 1.**

---

## Author Comment (AC5) · 14 Apr 2016

Dear Respective reviewer

Thank you very much for reviewing our manuscript and send us valuable comments.

---

## Author Comment (AC6) · 17 Apr 2016

**Vegetation Cover Change Detection and Assessment in Arid Environment Using Multi-temporal Remote Sensing images and Ecosystem Management Approach**

Anwar Abdelrahman Aly[1, 3], Abdulrasoul Mosa Al-Omran*[1], Abdulazeam Shahwan Sallam[1], Mohammad Ibrahim Al-Wabel[1], Mohammad Shayaa Al-Shayaa[2]

[1]Soil Science Dept., King Saud University, Riyadh, Saudi Arabia

[2]Agricultural Extension and Rural Community Dept., King Saud University, Riyadh, Saudi Arabia

[3]Soil and Water Science Dept., Faculty of Agric., Alexandria University, Egypt

*Corresponding Author: Tel: +966114678444; Fax: +966114678440

Email: rasoul@ksu.edu.sa; anwarsiwa@yahoo.com

**Abstract**

Vegetation cover (VC) changes detection is essential for a better understanding of the interactions and interrelationships between humans and their ecosystem. Remote sensing (RS) technology is one of the most beneficial tools to study spatial and temporal changes of VC. A case study has been conducted in the agro-ecosystem (AE) of Al-Kharj, in the centre of Saudi Arabia. Characteristics and dynamics of total VC changes during a period of 26 years (1987 - 2013) were investigated. A multi-temporal set of images was processed using Landsat images; Landsat4 TM 1987, Landsat7 ETM+ 2000, and Landsat8 to investigate the drivers responsible for the total VC pattern and changes which are linked to both natural and social processes. The analyses of the three satellite images concluded that the surface area of the total VC increased by 107.4% between 1987 and 2000, it was decreased by 27.5% between years 2000 and 2013. The field study, review of secondary data and community problem diagnosis using the participatory rural appraisal (PRA) method suggested that the drivers for this change are the deterioration and salinization of both soil and water resources. Ground truth data indicated that the deteriorated soils in the eastern part of the Al-Kharj AE are frequently subjected to sand dune encroachment; while the south-western part is frequently subjected to soil and groundwater salinization. The groundwater in the western part of the ecosystem is highly saline, with a salinity $\geq 6$ dS m$^{-1}$. The ecosystem management approach applied in this study can be used to alike AE worldwide.

**Keywords:** Change-detection, Remote sensing, Vegetation cover, PRA method, Al-Kharj agro-ecosystem

 **List of abbreviations**

(EM) ecosystem management; (RS) remote sensing; (GIS) geographic information systems; (GPS) global positioning systems, (LC) land cover; (LU) land use; (VC)  vegetation cover; (HA) holistic approach; (AE)

agro-ecosystem; (PRA) participatory rural appraisal; (ECw) electrical conductivity measured on groundwater samples; (ECe) electrical conductivity measured on soil samples using saturated paste extracts.

[revised manuscript text omitted]
 (Figure 3 a and b). The NDVI images could be classified into three classes, namely dense vegetation cover (NDVI > 0.5), moderate vegetation cover (NDVI 0.25 – 0.5), and sparse vegetation cover (NDVI < 0.25), as shown in Fig (4).

            **Figure 3.** NDVI classification for Landsat satellite image

               **Figure 4.** Vector layer for classified NDVI

 **2.5 Field Study**

**Water and Soil sampling and analysis**

A 180 groundwater samples were gathered from different locations in the Al-Kharj AE to cover the spatial variations of the ecosystem groundwater salinity (Fig. 5). All samples were analyzed for salinity using electrical conductivity (EC) meter (dS·m$^{-1}$) (Test kit Model 1500_20 Cole and Parmer) at 25 °C. The groundwater soluble calcium, magnesium, sodium, potassium, chloride, and sulfate were determined using

Ion Chromatography System (ICS 5000, Thermo (USA)); however, the bicarbonate and carbonate concentration were determined by titration with sulfuric acid ($H_2SO_4$) (Matiti, 2004). Furthermore, fifty soil samples were collected from studied area including deteriorated sites observed by satellite image for year

2013 (ground truth). A soil paste extract were prepared, and the ECe was measured for each sample (Klute,

1986). In addition, A 5TE (Decagon devices) soil moisture, EC, and temperature sensors were installed at three date palm field in the Al Kharj AE. The three Investigated fields have different changes of VC

between years 2000 and 2013. The first field (a) is located in eastern part of the study area with no change of VC and used fresh water for irrigation (ECw = 1.1 dS m$^{-1}$). The second (b) is a deteriorated field located in middle to the western part and used saline brackish water for irrigation (ECw = 6.5 dS m$^{-1}$). The third (c)

is abandoned field located in southern part of the study area with notable decrease of VC, this field has no irrigation activities due to the high salinity of groundwater (ECw=10.2 dS m$^{-1}$) (Fig 5). The first two irrigated fields adopted drip irrigation system.

**Coordinate & GIS Analysis**

In this study, the coordinates of the soils and groundwater samples were recorded by GPS with an accuracy of ~5 m. The GPS signal is corrected by a radio signal in real time. The locations of the ecosystem groundwater salinity (ECw) were configured as a comma-delimited text file (in the form of groundwater no, easting, and northing). The point data was overlaid on a satellite image by Arc GIS 9.3 software (ESRI,

2010) (Fig. 5). kriging interpolation, geostatistical method, of  ECw was carried out using kriging

                    **Figure 5.** Location of the studied wells interpolation tool of Geostatistical analyst in ArcGIS 9.3.

**Statistical analysis**

Statistical analysis was carried out using the statistical package for social sciences (IBM SPSS Statistics 21

Core System, IBM Corporation 2012). The statistical tests applied were basic statistics (maximum, minimum, mean, standard deviation, variance, standard error, median, skewness) and Spearman's correlation matrix (assuming $p < 0.01$).

**3. Results and discussion**

**3.1 Community Diagnosis of Ecosystem Problems**

Al-Kharj is a fragile ecosystem, highly vulnerable to environmentally induced land and water resources degradation. The ecosystem resource degradation problems in Al-Kharj are exacerbated by poor natural resource management and practices (Al-Omran et al., 2014).

The community diagnosis is considered a powerful investigation tool to overcome problems. The local people usually have an actual desire to solve their ecosystem problems. The PRA techniques were used to aid describe the issues that related to the characteristics of ecosystem to issues of agricultural and environment.

As an important tool of the PRA methodology, a **general village- hall meeting** was organized and held in the ecosystem. The farmers are most of ecosystem residents but some of the inhabitants have other employments e.g. merchants, civil servants and labors. The meeting was carried out by a large number of the ecosystem community and stakeholders' e.g. local government administrators, and engineer and staff of agricultural extension. The meeting was managed to recognize how the community understand and prioritize their land and environmental deteriorations and issues facing them (Aly, 2007, Chambers, 1994).

Agricultural problems summarized by community study

The PRA study found that the main agricultural problem in the studied ecosystem is the poor irrigation water quality which is causing soil salinization problems. This problem is aggravated by several factors such as:

- The numbers of new wells that were recently drilled for irrigation has increased dramatically causing depletion and deterioration of groundwater quality (Al-Omran et al., 2015; Aly et al., 2016).

- Poor irrigation practices in the Al-Kharj (excessive irrigation system). The same finding was recorded in Siwa AE, located in arid environment, by Aly et al. (2016)

- No agricultural drainage system in the Al-Kharj. Thus in some areas, the Al-Kharj could face the danger of water logging and salinization problems (Aly et al., 2016).

- Large investments in intensive cultivation which cultivates hundreds of acres, and drilled tens of new wells are causing great damage to the fragile ecosystem of the Al-Kharj (Algahtani et al., 2015; Aly, 2007; Sonneveld et al., 2016).

- In summer there is no agriculture activity due to high temperature (reached 50$^{o}$C) with exception protected area (Tourenq et al., 2009).

- Some farmer used desalination plant to overcome the irrigation water salinity (Al-Omran et al., 2014).

- Loss of biodiversity due to soil salinity. This is expected since salinization is the most common land degradation processes in arid and semi-arid regions (Farifteh et al., 2006)

- Farmers in Al-Kharj usually change their soils when deteriorated (Richardson et al., 2010).

- The rare and high cost of agricultural labor.

All the above mentioned agricultural issues and problems lead to a significant decrease in land productivity of the studied agro-ecosystem.

**3.2 Remote Sensing: Direction changes of vegetation cover**

The major changes detected in the study area between years 1987 and 2000 were the increasing of total VC in west and south-western part of Al-Kharj ecosystem (Table 1, Fig. 6). However, the total VC decreased between years 2000 and 2013 in the east and south-western part of Al-Kharj AE (Table 1, Fig. 6). The investigation of the three satellite images concluded that the surface area in square kilometers of the total VC increased dramatically between years 1987 and 2000 by 107.4%; however, it decreased by 27.5% between years 2000 and 2013 (Table 1) (Fichera et al., 2012). In an attempting to explicate the reason of the ecosystem total VC decrease in last decade, a relation between total VC and wheat production has been depicted. Figure (7) shows a direct relationship between wheat production in Saudi Arabia (USDA, 2015) and total VC in Al-Kharj AE. Furthermore, it recorded an evidence of progressive increase of wheat production and total VC during the period of 1984-1993 (Algahtani et al., 2015; Modaihsh et al., 2015; USDA, 2015). This was caused by the economic development that corresponds to the period of massive injection of subsidies that came with government's policy to expand the wheat production over this period (USDA, 2015). Rationally, this has led to a steady increase in the land area used up by vegetation. However, there were a nosedived during the period of 1994- 1998 due to the Saudi government stopped subsidies of wheat production to save water. A slight increase of total VC recorded between years 1998-2002, and a contentious decrease between years 2002- 2013. This study suggest that the decrease in the last decade of total VC was caused by land and water resources degradation (Sonneveld et al., 2016). This suggestion have been emphasized by field studies through PRA method and found in agreement with the findings of Algahtani et al. (2015).

**Table 1.** Spatio-temporal characteristics of Al-Kharj ecosystem

**Figure 6.** Change detection of vegetation cover

**Figure 7.** The changes of vegetation cover and wheat production

**3.3 Soil and water resources characteristics and its effects in agro-ecosystem**

The field study and observation, the review of secondary data, and community problem diagnosis using the PRA suggest that the driving role in the change of total VC recorded by RS in recent years are the soil and water resources deterioration and salinization. The ground truth found that the deteriorated soils are either subjected to salinization or sand dune encroachment (Figs 8, 9 and 10) (Al Omran et al., 2015; Alyemeni, 2000). In general, the sand dune in eastern part of studied AE is considered the main problem facing agriculture expansion; however, the groundwater salinity is considered the main problem of southwestern part (Figs 8, 9 and 10) (Al Omran et al., 2015).

Table (2) shows that in the eastern part of the ecosystem 83% of groundwater samples were suitable for irrigation with some restriction (ECw$\leq$ 3 dS m$^{-1}$) (Ayers and Westcot, 1985); however, the remaining their ECw ranged between 3-4 dS m$^{-1}$ (Table 2). In response to irrigation water salinity, 76% of irrigated soil ECe $\leq$ 4 dS m$^{-1}$, 18% ECe ranged between 4-10 dS m$^{-1}$, and 5% soil ECe >10 dS m$^{-1}$. Nonetheless, the VC area decreased by 18% between years 2000-2013. In the middle and western part, the ecosystem showed more vulnerable soil conditions for soil degradation. Only 64% of the groundwater can be considered suitable for irrigation (ECw$\leq$ 3 dS m$^{-1}$). However, 20% of groundwater samples ECw ranged between 3-4 dS m$^{-1}$, and 16% the ECw ranged between 4-10 dS m$^{-1}$. As a result, only 19% of the studied soil samples ECe$\leq$ 4 dS m$^{-1}$, 50% ECe between 4-10 dS m$^{-1}$, and regrettably 31% their ECe>10 dS m$^{-1}$. The VC is then decreased dramatically in this part by 33% between the years 2000-2013. The highest soil ECe in eastern part of studied ecosystem was 17.6 dS m$^{-1}$ (sample no 1); on the other hand, the middle part of the ecosystem deteriorated sites recorded 40.6 and 47.4 dS m$^{-1}$, samples no 17 and 18, respectively (Table 3 and Fig. 10). Moreover, the soil salinity dramatically increase in some sites of western ecosystem reaching 41.7 dS m$^{-1}$ (site no 29) (Figs 6 and 10). The groundwater in western part of studied ecosystem is considered highly saline since its salinity almost more than 6 dS m$^{-1}$ (Fig. 9). Mostly, no soil sodicity hazards are anticipated by using this type of groundwater in irrigation. The SAR of studied water samples were less than 10 with an average of 3.74 (Table 4) (Richards, 1954). In general, 34.8% of the arable land in the studied AE are considered saline (ECe > 4 dS m$^{-1}$), 34.8% are severely saline (ECe > 10 dS m$^{-1}$) and the remaining (30.4%) can be considered non saline (ECe < 4 dS m$^{-1}$). The ECe of Al-Kharj cultivated soils are ranged between 1 and 47.4 dS m$^{-1}$ for un-deteriorated and deteriorated sites, respectively; however, the uncultivated soil's ECe reached 140 dS m$^{-1}$ in western AE.

**3.4 Total Vegetation cover degradation and land and water resources salinity**

In order to prove that the land and water resources salinity of past ten years are the main cause of total VC

decrease in the ecosystem, the changes of total VC has been linked to water and soil salinity levels at three different fields (Fig. 11). The soil parameters (soil moisture, EC, and temperature) were recorded at investigated fields by sensors. The average values of soil parameters of four date palms at depth (0-30 cm)

for each field were presented (Fig. 11). The sensors in abandoned field did not work properly due to the low soil water content (~ 0.01 m$^3$m$^{-3}$) where the precipitation is negligible (Gao et al., 2014; Saha et al.,

2015). The results indicated that the irrigation with low water salinity in the first field did not lead to high soil salinity values (average soil's EC= 1.25 dS m$^{-1}$) (Fig. 11). The leaching process led to the soil salinity to get lower with adding irrigation. However, the irrigation with saline water in the second field led to soil quality deterioration due to salinity (average soil's salinity was equal to 6.7 dS m$^{-1}$) (Fig. 11). The soil in the abandoned field is suffering from severe salinity (averaged 39.2 dS m$^{-1}$) due to lack of irrigation and the low precipitation. Subsequently, soluble salts have been accumulated in the top soil layer negatively impacting on total VC water uptake and growth due to low tolerance of the total VC to very high salinity.

These are expected results as salinization and alkalinization are the most common land degradation

**Figure 8.** Sand dune encroachment

**Figure 9.** Interpolation of groundwater EC

**Figure 10.** Soil salinity in of studied ecosystem

**Table 2** Water and soil deteriorated parameter (salinity) and VC area

**Table 3** Statistical analysis of studied groundwater

**Table 4** Descriptive statistics of Al-Kharj groundwater processes in arid and semi-arid regions (Farifteh et al., 2006). Since the temperature of Al Kharj reaching 45 $^{\circ}$C in July, the soil temperature was also investigated in this study. Figure (11) clearly demonstrate that the summer irrigation led to dramatic decrease of soil temperature (up to 5 $^{\circ}$C). During the irrigation, the air is replaced with water leading to the decrease in soil temperature. On contrary, following the irrigation, the water drains and air would fill up the soil pores and the soil temperature gets higher (Fig. 11) (USAD, 2002). Comparing the three site VC, it is clear that the high salinity of the land caused by high salinity of groundwater resources had negative impact on vegetation survival especially in absence of leaching of salts by rainfall or fresh irrigation water. In addition, the sand dune encroachment represents another cause of the VC decrease in the eastern part of the study sites (Fig. 8). The farmers of Al-Kharj should be informed about the water quality of their wells and should be given advice by the extension services about the type of suitable crops and management that would safe guard the Al-Kharj ecosystem. The government should take an action to solve the problem of sand dune encroachment in the eastern part of the ecosystem, and help farmers to select salinity tolerance crops that can survive such conditions. Sand dune fixation is generally used to stop the dunes encroachment. Two methods are usually used; biological i.e., planting trees, shrubs and grasses species, and mechanical i.e., wooden sand fences and footpaths. Shelterbelt systems and afforestation, biological methods, using *Atriplex spp., Acacia spp, and Casuarina spp* were found efficient in stabilizing dunes in arid environment of Egypt, Senegal, and India (Draz et al., 1992; Kaul, 1985). In fact, the importance of the sand dunes fixation by afforestation is not only sand dune fixation but also can conserve arid ecosystem balance, and produce fuel and animals feed (Draz et al., 1992; Kaul, 1985).

In the USA, Tunisia, and Egypt saline waters have been successfully used for long irrigation time. The crops grown using this water are cotton, sugar beet, alfalfa, date palm, sorghum, barley, alfalfa, rye grass and artichokes (Rhoades et al., 1992). In Texas, USA, the saline groundwater (TDS = 2500 to 6000 mg/l) has been successfully used for three decades (Rhoades et al., 1992).

**Figure 11**. VC linked soil salinity

The suitability of saline groundwater for irrigation should be assessed for specific conditions including; crops type, soil characteristics, irrigation methods, cultural practices, and climatic conditions (Minhas,

1996). Many rational management option of saline irrigation water have been currently in use, some of them are: cyclic strategy, which involves using non-saline water and saline water in a repeating sequence, blending strategy which involves blending (dilution process) fresh with saline water, rotation strategy which means irrigation with low-salinity water for salt sensitive crops in a rotation with saline water for salt-tolerant crops  (Rhoades et al., 1992), planting salt tolerant crop varieties or genotypes / cultivars i.e., amaranth and quinoa which can survive under harsh conditions (Fghire et al., 2015; Pulvento, et al., 2015), and finally the use of computer model for assessing water suitability for crops production (Aly et al., 2015).

**4. Conclusions**

A comprehensive analyses of Al-Kharj, Saudi Arabia, agro-ecosystem components (physical resources and community) were conducted in this study. The field study and community-based diagnosis in addition to the use of satellite images to detect agriculture land-use changes over the twenty six years revealed that the groundwater and agricultural lands have been seriously degraded due to salinization. The major ecosystem changes detected by RS was total VC surface area increased between years 1987 and 2000 by 107.4%; however, it decreased by 27.5% between years 2000 and 2013. Between years 1984 and 1998, a direct relationship between wheat production in Saudi Arabia and total VC changes in studied AE is recorded.

The Saudi government subsidies to wheat production is governed the total VC changes in this period.

However, in the following years, the degradation of land and water resources induced the total VC changes.

This study found that the sand dune encroachment in eastern part of the AE is the main problem facing agriculture expansion; however, the land and groundwater salinity is considered the main problem in the middle and southwestern ecosystem. In the eastern ecosystem, 83% of the studied groundwater samples were suitable for irrigation with some restrictions (ECw$\leq$ 3 dS m$^{-1}$) and 76% of irrigated soil's ECe $\leq$ 4 dS

m$^{-1}$. However, in the middle and western part, 64% of the groundwater can be considered suitable for irrigation (ECw$\leq$ 3 dS m$^{-1}$), and only 19% of the studied soil samples its ECe$\leq$ 4 dS m$^{-1}$. The farmers of Al-

Kharj should be informed about the water quality of their wells and should be given advice by the extension services about the type of suitable crops and management that would safe guard the Al-Kharj ecosystem.

**Acknowledgment**

*This project was supported by NSTIP strategic technologies program number (12-ENV2581-02) in the*

*Kingdom of Saudi Arabia*

[revised manuscript text omitted]

[1] Med. = Median   [2] St.Dev. = Standard deviation

**Table 4.** Statistical analysis of groundwater chemical composition of Al-Kharj (n=180)

| | PH | EC dS m$^{-1}$ | Ca$^{2+}$ | Mg$^{2+}$ | Na$^+$ | K$^+$ | Cl$^-$ | HCO$_3^-$ | CO$_3^{-2}$ | SO$_4^{-2}$ | SAR |
| | | | | | | | meq L$^{-1}$ | | | | |
|---|---|---|---|---|---|---|---|---|---|---|---|
| Max. | 8.60 | 10.15 | 36.75 | 29.85 | 43.40 | 0.72 | 58.17 | 18.83 | 4.33 | 43.19 | 9.14 |
| Min. | 6.78 | 1.05 | 3.45 | 0.79 | 2.24 | 0.05 | 3.13 | 0.87 | 0.00 | 3.22 | 1.08 |
| Mean | 7.72 | 3.00 | 10.79 | 7.78 | 11.28 | 0.25 | 10.86 | 3.99 | 0.13 | 15.03 | 3.74 |
| St.Dev | 0.44 | 1.29 | 5.09 | 3.93 | 5.96 | 0.10 | 7.32 | 1.49 | 0.37 | 7.05 | 1.47 |
| Vari.[1] | 0.66 | 1.13 | 2.26 | 1.98 | 2.44 | 0.31 | 2.71 | 1.22 | 0.61 | 2.66 | 1.21 |
| St. error[2] | 0.18 | 0.23 | 0.33 | 0.31 | 0.34 | 0.12 | 0.36 | 0.24 | 0.17 | 0.36 | 0.24 |
| Med. | 7.72 | 2.64 | 9.60 | 6.69 | 10.21 | 0.23 | 9.50 | 3.83 | 0.00 | 12.83 | 3.51 |
| Skew.[3] | -0.15 | 2.47 | 1.39 | 2.16 | 2.53 | 1.66 | 3.85 | 5.96 | 8.20 | 1.18 | 1.12 |

[1]Vari. = Variance   [2]St. error = Standard error   [3] Skew. = Skewness